# How Many Faces Does the Plant U-Box E3 Ligase Have?

**DOI:** 10.3390/ijms23042285

**Published:** 2022-02-18

**Authors:** Xinguo Mao, Chunmei Yu, Long Li, Min Wang, Lili Yang, Yining Zhang, Yanfei Zhang, Jingyi Wang, Chaonan Li, Matthew Paul Reynolds, Ruilian Jing

**Affiliations:** 1The National Key Facility for Crop Gene Resources and Genetic Improvement, Institute of Crop Sciences, Chinese Academy of Agricultural Sciences, Beijing 100081, China; yuchunmei272@163.com (C.Y.); lilong01@caas.cn (L.L.); wanming0127@163.com (M.W.); 19956730179@163.com (L.Y.); zyn18845111029@163.com (Y.Z.); yfz15838298834@163.com (Y.Z.); wangjingyi@caas.cn (J.W.); lichaonan@caas.cn (C.L.); jingruilian@caas.cn (R.J.); 2International Maize and Wheat Improvement Center, Texcoco 56237, Mexico

**Keywords:** abiotic stress, biotic stress, development, E3 ubiqutin ligase, plant U-box protein, ubiquitination

## Abstract

Ubiquitination is a major type of post-translational modification of proteins in eukaryotes. The plant U-Box (PUB) E3 ligase is the smallest family in the E3 ligase superfamily, but plays a variety of essential roles in plant growth, development and response to diverse environmental stresses. Hence, PUBs are potential gene resources for developing climate-resilient crops. However, there is a lack of review of the latest advances to fully understand the powerful gene family. To bridge the gap and facilitate its use in future crop breeding, we comprehensively summarize the recent progress of the PUB family, including gene evolution, classification, biological functions, and multifarious regulatory mechanisms in plants.

## 1. Introduction

Ubiquitination is a major type of post-translational modification of proteins, occurring widely in eukaryotic cells, and plays pivotal roles in regulating the synthesis and degradation of proteins in plant growth, development and responses to environmental signals. Ubiquitin is a stable, highly conserved, and universally expressed protein. It covalently attaches to a lysine residue of target proteins, and therefore alters their stability, activity, trafficking and cellular localization. The ubiquitination system consists of several enzymes that act in concert. A ubiquitin-activating enzyme 1 (E1) catalyzes the formation of a thioester bond between E1 and ubiquitin in the presence of ATP and then delivers the activated ubiquitin to a ubiquitin-conjugating enzyme (E2); subsequently E2 transfers the ubiquitin to a ubiquitin-protein ligase (E3). E3 ligase plays a central role in determining the specificity of the ubiquitination system by candidate protein selection [1]. Eventually, the ubiquitin-labeled protein will be degraded by the ubiquitin 26S-proteasome system (UPS).

Given myriad downstream substrates, E3s are the most diverse components in the ubiquitination pathway [2]. For instance, more than 1300 E3 genes are identified in the Arabidopsis genome [3]. Based on subunit compositions and regulatory mechanisms, the E3s are divided into four main families, including Homologous to E6-associated protein Carboxyl Terminus (HECT), U-box, Really Interesting New Gene (RING) and Cullin-RING Ligase (CRL) [4]. Among them, the U-box type E3 ligase is the smallest family. 

## 2. Classification and Evolutionary Characteristics of U-Box Proteins

### 2.1. Structural Characteristics of the Conserved U-Box Domains

The U-box is a more recently identified protein domain with E3 ligase activity, which was first found in ubiquitin fusion degradation protein 2 (UFD2) in yeast [5,6]. The U-box domain is a modified RING-finger domain, lacking the zinc-chelating cysteine and histidine residues of RING-finger domains [7]. The conserved zinc-binding sites (cysteine and histidine residues) supporting the cross-brace arrangement in RING-finger domains are replaced by hydrogen-bonding networks in the U-box. The U-box scaffold is stabilized by a system of salt bridges and hydrogen bonds. The charged and polar residues participating in the network of bonds are more conserved than those in classic RING fingers, revealing their irreplaceable role in maintaining the stability of U-box [7]. A conservative Val→Ile substitution disrupts structural integrity of the U-box and leads to pre-mRNA splicing defects of Prp19p [8]. Moreover, both U-box and RING fingers contain a conserved interaction surface. Mutagenesis of residues at the interface does not perturb the structure of U-box, while abrogating the function of Prp19p in vivo [8]. In Arabidopsis, identical amino acids are present in the corresponding conservative positions in the U-box domains of AtPUB14 and many other plant U-box proteins [9].

### 2.2. Evolutionary Characteristics of PUBs in the Viridiplantae

The number of U-box proteins varies greatly in fungi, animals and plants. There are only two U-box genes, i.e., *UFD2* and *PRP19*, in the yeast genome [6]. In humans, eight U-box genes are found, including *UBE4A*, *UBE4B*, *UIP5*, *PRP19*, *CHIP*, *CYC4*, *WDSUB* and *Act1* [10]. Intriguingly, six of the seven main U-box genes of humans (*UBE4A*, *UBE4B, UIP5, PRP19, CHIP* and *CYC4*) are present in the ancestor of all current metazoans, and *WDSUB* is found in placozoans, cnidarians and bilaterians, revealing that the U-box genes are evolutionarily conserved and occurred before the emergence of animals [10]. 

In Viridiplantae, including green algae, the plant U-box (PUB) E3 ligase gene family has expanded remarkably compared to those of fungi and animals. For instance, there are 30, 64, 67, 62 and 77 members in the genomes of *Chlamydomonas reinhardtii* [11], Arabidopsis [12], barley [13], tomato [14] and rice [2], respectively. For polyploidy plant species, the gene number is even greater because of the occurrence of whole genome duplication [15,16,17]. For instance, over 200 PUB genes are present in the genome of tetraploid cotton and hexaploid wheat [15,16]. Phylogenetic and gene structure data have shown that the PUB gene families have experienced various duplication events. In *Chlamydomonas*, one gene duplication event is found [11]. In barley (*Hordeum vulgare* L.), two or more *HvPUB* genes are arranged tandemly or closely clustered together on the chromosomes. For instance, *HvPUB11/12* and *HvPUB58/59* are in tandem and have 100% sequence identity, indicating that gene duplications occurred recently. However, some gene pairs, such as *HvPUB6/43/52*, *HvPUB13/25*, *HvPUB15/16* and *HvPUB28/29*, are different, hinting that the duplication events might be occurred at earlier time [13]. In *Gossypium raimondii*, 15 segmental duplications and three tandem duplications are found in the homologous gene pairs, and 25 segmental duplications and two tandem duplications are identified in *G. arboretum* [16]. In tetraploid cotton *G. hirsutum*, more gene pairs are identified than the sum of its genome donors, suggesting that the duplication events happened post-polyploidization [16]. In wheat, 18 pairs of duplicated U-box E3 genes are identified, and six pairs show duplication events in the equal chromosome on 4A, 5A, 5B, and 6B, 12 pairs of segmental duplication events are detected between different chromosomes, such as between chromosomes 4A and 7A/7D, 4B/4D and 5A, and 5A/5B/5D and 6A [15]. It is hypothesized that the wide expansion of PUB genes is an essential strategy for sessile plants to adapt the environmental challenges [18], very much in tune with their functions in responses to biotic and abiotic stresses.

### 2.3. Classifications of PUBs in Various Plant Species 

To characterize U-box containing proteins, the protein family is further divided into several groups based on the presence or absence of distinctive functional domains, such as ARM (Armadilo repeat), CYC4 (cyclophilin), GKL motif, UFD2, U-box, WD40, TRR (tetratrico-peptide repeat), and protein kinase domain [2,13]. The PUBs are divided into three groups in Chlamydomonas; five in pear; six in banana; seven in wheat, peach and cabbage; eight in Arabidopsis, rice, barley and soybean (Table 1). However, there is no one-to-one correspondence between the classifications from various plant species, because of the huge differences in gene number and domain/motif types, as well as the classification criteria employed by researchers. Moreover, the emergence of new PUB proteins further complicates the classifications [15,16]. 

## 3. PUBs Are Involved in the Regulation of Gene Expression in Different Ways

### 3.1. PUBs Control the miRNA Biogenesis

MicroRNAs (miRNAs) are endogenous small non-coding RNAs controlling gene expression by guiding the cleavage or translational inhibition of complementary mRNAs. Compelling evidence shows that miRNA participates in various biological processes in plants. The biogenesis of miRNA in plants requires numerous proteins, and the processing complex consists of the core components DICER-LIKE 1 (DCL1), SERRATE (SE) and HYPONASTIC LEAVES (HYL1). Among proteins associated with the DCL1 complex, CDC5 and PRL1 are two core subunits of the conserved MOS4-associated complex (MAC) [24]. MAC contains three core subunits, MAC3A, MAC3B, and MOS4, and more than 13 accessory proteins with diversified functions. MAC3A and MAC3B are conserved U-box-containing proteins in eukaryotes. Deficiency in MAC impairs plant immunity and development [24]. Mutation of both MAC3A and MAC3B leads to the reduction of miRNA and primary miRNA transcripts (pri-miRNAs) levels, and causes elevated transcripts of miRNA targets in Arabidopsis. In vivo evidence has shown that MAC3A associates with pri-miRNAs, indicating that MAC3A and MAC3B may stabilize pri-miRNAs. Furthermore, MAC3A and MAC3B interact with the DCL1 complex that catalyzes miRNA maturation, promote DCL1 activity, and are required for the D-body localization of HYL1 [25]. MAC functions as a complex to control miRNA levels through modulating pri-miRNA transcription, processing and stability. However, MAC3A and MAC3B act redundantly in miRNA biogenesis [25].

### 3.2. PUBs Are Involved in the Signal Transductions of Various Phytohormones

Compelling data have established that PUBs are involved in the regulating gene expression via different phytohormone signaling pathways. Brassinosteroid-Insensitive 1 (BRI1) is a leucine-rich repeat receptor-like kinase that functions as the cell surface receptor for brassinosteroids (BRs), transducing the extracellular BR signal into the nucleus. BR perception promotes BRI1 ubiquitination by AtPUB12 and AtPUB13, and association with AtPUB12 and AtPUB13 through phosphorylation at serine 344 residue. Loss of AtPUB12 and AtPUB13 results in reduced BRI1 ubiquitination and internalization accompanied with a prolonged BRI1 plasma membrane-residence time, suggesting that the ubiquitination of BRI1 by AtPUB12/13 is a key step in BRI1 endocytosis [26].

In rice, *Taihu Dwarf 1* (*TUD1*) encodes a functional PUB E3 ubiquitin ligase. *D1* encodes the α subunit of heterotrimeric G-protein RGA1, and plays important roles in many signal transduction pathways. Genetic, phenotypic, and physiological data have shown that OsTUD1 is epistatic to OsD1/RGA1 and the *ostud1* mutant is less sensitive to BR treatment. Histological observations show that the dwarf phenotype of *ostud1* is mainly due to decreased cell proliferation and disorganized cell files in aerial organs. Protein interaction assays show that OsTUD1 directly interacts with OsD1/RGA1, demonstrating that OsTUD1 and OsD1 act together, therefore influencing plant growth and development via an OsD1/RGA1-mediated BR-signaling pathway in rice [27].

*PHOR1* (photoperiod-responsive 1), which encodes an ARM repeat-containing PUB protein in potato (*Solanum. tuberosum* ssp. *andigena*), is required in the process of tuberization under short-day conditions. Antisense inhibition of *PHOR1* produces a semi-dwarf phenotype similar to that of gibberellic acid (GA)-deficient plants, and the antisense lines show reduced GA responsiveness combined with a higher endogenous GA content than the WT plants. Conversely, transgenic lines overexpressing *PHOR1* display an enhanced response to GA. Application of exogenous GA induces a rapid migration of PHOR1-GFP protein to the nucleus. Hence, PHOR1 is a key component of GA signaling pathway [28,29].

MYC2 is an important regulator for jasmonic acid (JA) signaling. AtPUB10 is an ARM repeat-containing PUB. Both MYC2 and AtPUB10 are nucleus localized. AtPUB10 interacts with and ubiquitinates MYC2 jointly with E2 enzyme UBC8. MYC2 is unstable in WT plants, whereas its stability is enhanced in *atpub10* mutant, suggesting the destabilization by AtPUB10 [30]. Furthermore, the half-life of MYC2 is shortened in *AtPUB10* plants, but prolonged in *atpub10* mutant and the dominant-negative AtPUB10 (C249A) mutant. Root growth of *atpub10* seedlings phenocopies MYC2-overexpressing seedlings and is hypersensitive to methyl jasmonate, whereas *AtPUB10* transgenic plants and *jin1-9* (*myc2*) seedlings are hyposensitive JA. In addition, the root phenotype conferred by MYC2 overexpression in double transgenic plants is reversed or enhanced by induced expression of *AtPUB10* or *AtPUB10 (C249A)*, respectively. In short, AtPUB10 participates in the JA signaling pathway by targeting MYC2 for degradation [30].

The S-domain receptor kinases (SRKs) are positive regulators of self-incompatibility (SI) response, and capable of phosphorylating the ARM repeat domains of PUBs in vitro. ARK1 (SD1-7) is a member of SRKs in Arabidopsis. AtPUB9 is a U-box protein with ARM domains. Both ARK1 and AtPUB9 display a predominant localization in the nucleus, a steady-state localization pattern [31]. Interestingly, AtPUB9 exhibits redistribution to the plasma membrane of tobacco BY-2 cells when either treated with ABA or co-expression with the active kinase domain of ARK1. As well, T-DNA insertion mutants for ARK1 and AtPUB9 lines show altered sensitivity to ABA during seed germination. Genetic assays reveal that ARK1 and AtPUB9 most likely functions in a linear fashion, and act at or upstream of ABI3, demonstrating their potential involvements in ABA responses [31].

Additionally, PUBs are also implicated in various phytohormone-mediated pathways of fruit development and ripening (see the section of fruit ripening). 

## 4. PUBs Are Omnipotent Players in the Responses to Various Abiotic Stress

Expression profile is the most direct indicator of involvement of a gene in a specific biological process. Fueled by the development of genome sequencing and the popularity of transcriptome sequencing, many PUB genes are identified as participants in the response to abiotic stress. For instance, a total of 118 PUB genes show drought- and heat-stress-specific expression patterns in wheat [15], 60 genes are induced by drought/salt/low temperature in banana [20], 11 out of 67 *HvPUB* genes are at least two-fold induced by drought in barley seedlings [13], supporting the extensive involvements of PUBs in abiotic stress responses.

### 4.1. PUBs Are Pivotal Regulators in the Response to Drought Stresses

Drought stress is the major environmental limitation for plant growth and development. Many PUBs are involved in the response to drought stress and serve as positive or negative regulators in the processes. In Arabidopsis, several PUBs play negative roles in response to drought stress (Table 2). For instance, AtPUB11 negatively modulates drought tolerance by degrading ABA and drought stress responsive receptor-like protein kinases, Leucine-Rich Repeat Protein 1 (LRR1) and kinase 7 (KIN7). Mutation of *AtPUB11* leads to a phenotype of drought tolerance, while *lrr1* and *kin7* mutants are more sensitive to drought stress than WT [32]. AtPUB18 and AtPUB19 are homologous PUB E3 ligases. The *atpub18-2atpub19-3* double mutant displays more sensitivity to ABA and enhanced drought tolerance than each single mutant plants and WT [33]. AtPUB22 and AtPUB23 coordinately control a drought signaling pathway by ubiquitinating cytosolic RPN12a in an ABA-independent manner. In this process, the two PUBs act as negative regulators in response to drought stress [34]. The *atpub18-2atpub19-3atpub22atpub23* quadruple mutant exhibits enhanced tolerance to drought stress as compared with each *atpub18-2atpub19-3* and *atpub22atpub23* double mutant progeny; however, the stomatal behavior of the quadruple is almost identical to the *atpub18-2atpub19-3* double mutant in the presence of ABA, H_2_O_2_, and calcium [33]. A major structural difference between AtPUB18/AtPUB19 and AtPUB22/AtPUB233 is the presence or absence of the U-box N-terminal domain (UND), which is a determinant of substrate specificity. UNDPUB18 is critical in the negative regulation of ABA-mediated stomatal movements, and Exo70B1, a subunit of the exocyst complex, is a target of AtPUB18, whereas Exo70B2 is a substrate of AtPUB22 [35]. AtUBC32, AtUBC33, and AtUBC34 comprise group XIV E2 ubiquitin-conjugating enzymes, are co-localized with AtPUB19 to the punctate-like structures in Arabidopsis. Suppression of AtUBC32, AtUBC33, and AtUBC34 results in increased ABA-mediated stomatal closure and strengthened tolerance to drought stress [36].

In rice, OsPUB41 and OsPUB67 play different roles in responding to drought. OsPUB41 is a cytosolic and nuclear localized PUB E3 ligase. *OsPUB41* is specifically induced by dehydration and ABA treatments. The core U-box motif of OsPUB41 possesses the E3 ligase activity and can be activated by OsUBC25 [37]. The RNAi knockdown and *ospub41* mutation plants exhibit enhanced tolerance to drought stress compared with the WT plants in terms of transpirational water loss, long-term dehydration response, and chlorophyll contents. The chloride channel protein OsCLC6 is a putative substrate of OsPUB41. Hence, OsPUB41 acts as a negative regulator of dehydration by interacting with a candidate substrate OsCLC6 [37]. *OsPUB67* is significantly induced by drought, salt, cold, JA, and ABA, and expressed in the nuclei, cytoplasm, and membrane systems. OsPUB67 interacts with two drought tolerance negative regulators OsRZFP34 and OsDIS1 on the stomata, and improves drought tolerance by enhancing the ability of reactive oxygen (ROS) scavenging and stomatal closure [38]. Transcriptomic assays reveal that OsPUB67 participates in regulating the expressions of abiotic stress responsive genes in an ABA-dependent manner [38].

In *Glycine max*, 125 PUB genes are identified [17]. Among them, nine PUB proteins, GmPUB1–GmPUB 9, are involved in the response to water deficit. GmPUB6 is a peroxisome localized E3 ubiquitin ligase [39], while GmPUB8 is localized to post-Golgi compartments, interacting with GmE2 protein [17]. Both are induced by ABA, high salinity and water deficit. Overexpression of *GmPUB6* and *GmPUB8* results in decreased plant survival rates, reduced seed germination, retarded plant growth under osmotic stress, and suppressed ABA- or mannitol-mediated stomatal closure, relative to the WT control. Moreover, multiple stress responsive genes, including *ABI1*, *DREB2A*, *KIN2*, *RAB18*, *RD20*, *RD29A* and *RD29B*, are suppressed in transgenic plants under dehydration conditions [17,39]. 

TaPUB1 is a key regulator in the response to multiple adverse environmental stimuli in wheat. Constitutive expression of *TaPUB1* in *Nicotiana benthamiana* leads to enhanced tolerance to water deficit relative to WT, which is verified by several improved morphological and physiological traits in transgenic lines, including higher seed germination, increased survival rates, strengthened photosynthetic, water retention and antioxidant abilities, and less ROS accumulation [42]. 

StPUB27 is a negative modulator of drought tolerance in potato. Overexpression *StPUB27* causes increased stomatal conductance and results in accelerated water loss of detached leaves compare to the non-transgenic control. However, the RNA interference plants show a phenotype of smaller stomatal conductance and higher water retention capability, indicating that StPUB27 negatively regulate the response to drought stress by controlling stomata aperture in potato [32].

CaPUB1 is a PUB member in hot pepper (*Capsicum annuum* L.). Ectopic expression of *CaPUB1* in Arabidopsis and rice results in hypersensitive phenotypes to drought stress, with lower survival rates and chlorophyll contents than WT [40,41]. Furthermore, the expression of *RD29a*, a typical drought-induced gene, is substantially suppressed in transgenic Arabidopsis compared to the WT plants [41]. Hence, CaPUB1 functions as a negative regulator in the response to drought stress in both Arabidopsis and Rice [40,41].

*PalPUB79* is significantly induced by drought, salinity and ABA signaling in *Populus alba* L. *PalPUB79* overexpression confers enhanced drought tolerance in transgenic poplars, and this phenotype is eliminated in the absence of ABA signaling. However, *PalPUB79* RNAi lines are more sensitive to drought stress compared to WT [46]. Furthermore, PalPUB79 interacts with PalWRKY77, a negative transcriptional regulator (TF) of ABA signaling, and mediates its ubiquitination for degradation, thus neutralizing the inhibitory effect on the expression of *PalRD26*, a drought responsive marker gene. In turn, PalWYKY77 directly down-regulates *PalPUB79* expression by binding to the W-box (PW2) in the *PalPUB79* promoter [46]. However, the inhibition can be reversed by adding PalPUB79 in dual-luciferase assay systems, suggesting that a negative feedback loop between PalWRKY77 and PalPUB79 during ABA signaling in poplar [46].

PbrPUB18 is a nucleus-localized PUB protein in pear (*Pyrus bretschneideri* L.). Heterologous expression of *PbrPUB18* confers enhanced drought tolerance in Arabidopsis, manifested by light leaf-wilting symptoms and a serial of improved physiological traits related to drought tolerance, including enhanced photosynthetic ability, strengthened cell membrane stability, less accumulation of malondialdehyde and ROS. Thus, PbrPUB18 plays a positive role in the response to water deficit [23].

### 4.2. PUBs Are Key Components in the Response to Salt Stress

High soil salinity is one of major limitations for crop yield and production. Approximately 40% of irrigated lands worldwide are affected by increased salt levels, and the expansion of soil salinization is a serious threat to crop performance [56]. Solid data have shown that PUB E3 ligases are indispensable components in salt stress responses in plants. 

PnSAG1 is an ARM-domain containing PUB in Antarctic moss *Pohlia nutans*, which is rapidly induced by exogenous ABA, salt, cold and drought stresses [47]. The *PnSAG1-overexpressing* Arabidopsis lines exhibit more sensitive to ABA and salt stress during seed germination and early root growth. Similarly, heterogeneous expression of *PnSAG1* in *Physcomitrella patens* causes hypersensitive to ABA and high salinity in gametophyte growth [47]. Gene expression assays show that the expression of salt stress/ABA-related genes are dramatically down-regulated in *PnSAG1* transgenic plants under salt stressed conditions. Therefore, PnSAG1 plays negative roles in the responses to ABA and salt treatments [47].

In Arabidopsis, AtPUB30 participates in salt stress response at seed germination stage in an ABA-independent manner [57]. The *Atpub30* mutant shows more tolerance to salt stress in seed germination relative to WT, whereas the mutant of its closest homolog *AtPUB31* shows mild sensitivity to salt stress. AtPUB30 specifically interacts with and ubiquitinates BRI1 kinase inhibitor 1 (BKI1), a regulator playing dual roles in BR signaling, and degrades BKI1 via UPS. The *bki1* mutant was sensitive to salt, whereas the *BKI1* transgenic plants present a salt-tolerant phenotype. In sum, *AtPUB30* negatively regulates salt tolerance probably through regulating the degradation of BKI1 in BR-signaling pathway [48].

Several wheat PUBs are identified as salt responsive regulators. *TaPUB1* overexpression results in the up-regulation of ion channel genes in rice, tobacco and wheat under high salinity conditions, causing substantial increase of net root Na^+^ efflux and decreases in net K^+^ efflux and H^+^ influx, therefore maintaining lower cytosolic Na^+^/K^+^ ratios in transgenic plants relative to the non-transgenic plants [42,43,44,49]. Furthermore, *TaPUB1* promotes the expression of some salt responsive genes and enhances the antioxidant capacity under salt stress. Consistent with this, the RNAi-mediated knockdown plants show an opposite phenotype to salt stress. Moreover, TaPUB1 interacts with TaMP (α-mannosidase protein), an important regulator of salt response in both yeast cells and plants. In short, TaPUB1 positively regulates salt tolerance by interacting with α-mannosidase in plants [42,43,44,49]. 

TaPUB15 is an ortholog of *OsPUB15* and induced by salt, ABA, low temperature, and water deficit. Overexpression of *TaPUB15* causes substantial up-regulations of several salt induced genes and decreased Na^+^/K^+^ ratios relative to WT, and therefore confers pronounced salt tolerance in transgenic rice. These results are also verified in transgenic Arabidopsis, demonstrating TaPUB15 plays a role in enhancing salt tolerance in both monocot and dicot species [49]. *TaPUB26* is induced by high salinity, cold, drought and phytohormones. Ectopic expression of *TaPUB26* in *Brachypodium distachyon* leads to a hypersensitive phenotype to salt stress, manifested by reduced chlorophyll contents, decreased photosynthetic capabilities and antioxidant enzyme activities, more ROS accumulation and severer cell membrane damage relative to the WT plants [50]. Moreover, the transgenic plants have higher Na^+^ contents and lower K^+^ contents following salt treatment, hence maintain a higher cytosolic Na^+^/K^+^ ratio in plant cells. Moreover, TaPUB26 interacted with TaRPT2a, an ATPase subunit of the 26S proteasome complex, suggesting that it might modulate salt response by interacting with and ubiquitinating TaRPT2a [50].

The hot pepper *CaPUB1* is also involved in the response to salt stress. Its overexpression causes a hypersensitive phenotype to salt stress in Arabidopsis. Both germination and post-germination growth of *CaPUB1* transgenic plants are severely inhibited by mild salinity, while the WT plants are only slightly suppressed. Moreover, WT and transgenic plants display similar sensitivity to exogenous ABA in seed germination, consistent with its irresponsive expression to ABA, suggesting that the negative response to salt stress mediated by CaPUB1 is ABA-independent [41]. 

### 4.3. PUBs Are Key Regulators in the Response to Extreme Temperatures

Extreme temperatures, including heat and cold/freezing, seriously affects plant growth, development and geographical distributions. Increasing data have shown that PUB proteins are implicated in the response to extreme temperatures in different plant species (Table 2).

AtPUB48 is a nucleus-localized PUB E3 ligase. Its overexpression leads to enhanced thermotolerance in seed germination and seedling growth in Arabidopsis, with remarkably enhanced transcription of several heat responsive genes, including *HSP101*, *HSP70*, *HSP25.3*, *HSFA2*, and *ZAT12*. In line with this, disruption of *AtPUB48* causes a reduced germination rate relative to WT under high temperature conditions, and the expression levels of the heat responsive genes are dramatically reduced. Therefore, AtPUB48 may target the unknown substrate receptor to 26S proteasome proteolysis [51].

CHIP (Carboxyl Terminus of the HSC70-Interacting Protein) is a type of conserved chaperone-dependent PUB targeting misfolded proteins. *SlCHIP* is induced by high temperature and stress hormones in tomato. Silencing of *SlCHIP* leads to a phenotype hypersensitive to high temperature, accompanied with reduced photosynthetic activity, elevated electrolyte leakage and accumulation of insoluble protein aggregates. The accumulated protein aggregates are highly ubiquitinated, hinting that other E3 ligases might be involved in the ubiquitination process. Overexpression of *SlCHIP* restores the phenotype of the *slchip* mutant under heat-stressed conditions. Thus, SlCHIP might modulate thermotolerance by targeting the degradation of misfolded proteins produced during heat stress [58].

Cold and freezing stresses adversely affect plant growth, development, and crop productivity and quality. AtPUB25 and AtPUB26 are two positive regulators of freezing tolerance in Arabidopsis. Both can poly-ubiquitinate AtMYB15, a transcriptional repressor of the CBF-dependent cold signaling pathway, leading to its degradation and thus enhancing CBF expression under cold stress. Furthermore, cold-activated OST1 can specifically phosphorylate AtPUB25 and AtPUB26 at conserved threonine residues, enhance their E3 ligase activity, and facilitate the cold-induced degradation of AtMYB15. Collectively, the AtOST1-AtPUB25/26 module regulates the duration and intensity of cold response by controlling the homeostasis of AtMYB15 [53].

VpPUB24 is a PUB member in the Chinese wild grapevine (*Vitis pseudoreticulata*), and implicated in the responses to several abiotic stresses, especially cold [55]. ICE1 (inducer of CBF expression 1) is an upstream TF regulating the expression of CBF genes under cold stress [54]. VpICE1 is targeted for degradation via the 26S proteasome, and the degradation is accelerated by VpHOS1, but not by VpPUB24. Immunoblot data show that VpPUB24 promotes the accumulation of VpICE1, and suppresses the expression of *VpHOS1* to regulate the abundance of VpICE1 [55]. Furthermore, VpICE1 promotes the transcription of *VpPUB24* at low temperatures. Moreover, VpPUB24 interacts with VpHOS1 in yeast cells, hinting that VpPUB24 might ubiquitinate VpHOS1. Overexpression of *VpPUB24* confers enhanced cold tolerance in Arabidopsis. Taken together, VpPUB24, VpICE1 and VpHOS1 form a complex module to regulate cold stress response [55].

CaPUB1 is a multifaceted PUB E3 ligase in hot pepper. The *CaPUB1* transgenic rice lines present a tolerant phenotype to prolonged cold stress compared with the WT plants, with higher survival rates and chlorophyll contents, and less electrolyte leakage. Furthermore, several cold stress-induced marker genes, including *DREB1A*, *DREB1B*, *DREB1C*, and *Cytochrome P450*, are significantly induced by cold stress in transgenic lines than in the WT plants. Hence, CaPUB1 acts as a positive regulator of cold stress in transgenic rice [40]

### 4.4. TaPUB1 Is an Essential Regulator of Heavy Metal Tolerance

Heavy metals, including cadmium, silver and mercury, dramatically affect plant growth and development. Heavy metals in the soil are absorbed by plant roots and accumulate in aerial tissues, which seriously retards several molecular and physiological processes. Cd^2+^ is one of the most deleterious heavy metal pollutants. Recent progress shows that TaPUB1 is implicated in regulating Cd^2+^ absorption in wheat. The transcription of *TaPUB1* is significantly induced by Cd stress and indole acetic acid (IAA) treatment [45]. Wheat plants overexpressing *TaPUB1* show significantly reduced Cd^2+^ uptake and accumulation, whereas RNAi plants exhibited a substantial increase in Cd^2+^ accumulation following Cd treatment. Protein interaction results show that TaPUB1 interacts with and ubiquitinates TaIRT1, a Cd^2+^ transporter, resulting in the inhibition of Cd^2+^ uptake, decreased ROS accumulation and down-regulated antioxidant enzyme activities under Cd-stressed conditions. Furthermore, TaPUB1 directly interacts with and ubiquitinates TaIAA17, facilitating its degradation, thus releases the inhibition on roots and fuels primary root elongation by activating the auxin signaling pathway under Cd stress [45].

### 4.5. PUBs Are Implicated in the Response to Nitrogen Starvation

Nitrogen is an important nutrient element that influences lipid/carbohydrate accumulation in microalgae [59]. In total of 30 *CrPUB* genes, 25 are involved in the response to nitrogen starvation in *C. reinhardtii*. Among them, 18 *CrPUBs* are induced by N-deficit, and seven are inhibited, revealing that CrPUBs are widely involved in the response to N-starvation. Furthermore, silencing of *CrPUB5* and *CrPU14* causes significant lipid accumulations under N-deplete conditions, whereas knockdown of *CrPUB11*, *CrPUB23* and *CrPUB28* results in substantial reduction of lipid accumulation, indicating the intensive involvements of CrPUBs in oil metabolism under N-starvation [11].

## 5. PUB E3 Ligases Are Pivotal Regulators in the Response to Biotic Stress

Plants are constantly exposed to a variety of pathogen factors. To survive plants have developed numerous mechanisms to cope with the challenges. PUB E3 ligases are key components of diverse signaling pathways in the responses to pathogen attacks.

In Arabidopsis, several PUB proteins have been identified as immunity modulators (Table 3). Convincing data show that the closely related AtPUB12 and AtPUB13 are involved in response to pathogen infections. The Arabidopsis pattern-recognition receptor Flagellin Sensing 2 (FLS2) recognizes bacterial flagellin and initiates immune signaling through association with BAK1. BAK1 phosphorylates AtPUB12 and AtPUB13 and is required for FLS2-PUB12/13 association. AtPUB12 and AtPUB13 polyubiquitinate FLS2 and promote flagellin-induced FLS2 degradation, which in turn attenuates FLS2 signaling to prevent excessive or prolonged activation of immune responses [26,60]. Moreover, AtPUB13 is also implicated in innate immunity [61,62]. Disruption of *AtPUB13* causes spontaneous cell death, accumulated of H_2_O_2_ and salicylic acid (SA), and enhanced resistance to biotrophic pathogens, but increased the susceptibility to necrotrophic pathogens. SID2 (SA-induction deficient 2) and PAD4 (Phytoalexin deficient 4) are required in SA-mediated disease resistance [61]. AtPUB25 and AtPUB26 negatively regulate the defense against hemibiotrophic pathogen *Verticillium dahlia* by degrading the positive regulator AtMYB6, which promotes plant resistance to Verticillium wilt. Mutation of AtPUB25 and AtPUB26 leads to strengthened resistance to Verticillium. VDAL is a *Verticillium dahliae*-secreted Asp f2-like protein and competes with AtMYB6 for binding to PUBs and reducing hypersensitive response (HR) caused by infection, thus keep host plant alive. Alternatively, hemibiotrophic pathogens may take nutrients from host cells [63].

In rice, several PUB E3 ligase are identified as critical regulators in disease resistance (Table 3). OsSPL11, an ortholog of AtPUB13, functions as a negative regulator of programmed cell death (PCD). Mutation of *spl11* confers enhanced non-race-specific resistance to both *Xanthomonas oryzae* and *Magnaporthe oryzae*. OsSPL11 physically interacts with and ubiquitinates the GAP protein SPIN6 to suppress NADPH oxidase-mediated ROS generation and PR gene activation, therefore to inhibit the autoactivation of defense responses. The resistance is correlated with a constitutive activation of defense-related genes, including pathogenesis-related (PR) genes (*PR1*, *PBZ1*, *chintinase III*), oxalate oxidase genes involved in the production of ROS (*HvOxOa*, *HvOxOLP*), and genes encoding peroxidases (*POX8.1*, *POX22.3*) [74]. Moreover, SPL11 also negatively regulates SA accumulation, which might inhibit SA-mediated resistance and lead to susceptible to pathogen factors [65]. OsPUB15 is a major component of the transmembrane receptor-like kinase Pid2 mediated signaling pathway. Pid2 can phosphorylate OsPUB15, and the phosphorylated OsPUB15 has E3 ligase activity. *OsPUB15* overexpressing rice plants display cell death lesions associated with constitutive activation of basal defense responses includes excessive accumulation of H_2_O_2_, up-regulation of disease-related genes, and increased resistance to *M. oryzae* spore strains [66]. OsPUB44 directly interacts with XopP (Xoo), an effector of pathogen *X. oryzae*. Silencing of *OsPUB44* suppresses peptidoglycan- and chitin-triggered immunity, and the plants exhibit a susceptible phenotype to *X. oryzae*, indicating that OsPUB44 positively regulate immune responses by degrading XopP via UPS [67].

P3IP1 (P3 Induced Protein 1) is named for being induced by the P3 protein of rice grassy stunt virus (RGSV). Stable expression of P3 protein in rice leads to developmental abnormities similar to the disease symptoms caused by RGSV, such as dwarfism and excess tillering. Both transgenic expression of P3 and RGSV infection induce ubiquitination and UPS-dependent degradation of rice Nuclear RNA Polymerase D1a (OsNRPD1a), one of two orthologs of the largest subunit of plant-specific RNA polymerase IV, which is required for RNA-directed DNA methylation. P3IP1 interacts with and ubiquitinates OsNRPD1a, and mediates its degradation via UPS in vitro and in vivo. Knockdown of *OsNRPD1* or overexpression of P3IP1 results in the phenotypes similar to RGSV disease symptoms in rice, thus P3IP1 negatively modulate the response to virus attacks [68].

StPUB17 is a nucleus-localized positive regulator of programmed cell death (PCD) triggered by resistance proteins CF4/9 in potato. Silencing *StPUB17* in potato by RNAi and *NbPUB17* in *N. benthamiana* by virus-induced gene silencing each enhanced the leaf colonization of *Phytophthora infestans*, because of attenuated PAMP-triggered immunity (PTI). Further data reveal that not all PTI- and PCD-associated responses require PUB17 [69]. Exclusion of the StPUB17 (V314I, V316I) mutant from the nucleus abolishes its dominant-negative activity, demonstrating that StPUB17 functions in the nucleus. Overall, StPUB17 is a positive regulator of immunity to late blight, acting in the nucleus to promote specific PTI and PCD pathways [69]. Ectopic expression of the Arabidopsis ortholog *AtPUB17* in *ACRE276*-silenced tobacco plants rescues HR. The *atpub17* mutant also displays decreased resistance to avirulent *Pseudomonas syringae* pv. tomato [75]. *GhPUB17*, an ortholog of *StPUB17*, is induced by the infection of *V. dahliae* or exogenous hormone treatment, including JA and SA in cotton. *GhPUB17*-knockdown cotton plants are more resistant to *V. dahliae*, conversely the overexpressing plants are more susceptible to the pathogen, indicating that GhPUB17 is a negative regulator of cotton resistance to *V. dahlia*. Cyclophilin protein GhCyP3 is an interactor of GhPUB17 with antifungal activity. The ubiquitination activity of GhPUB17 is inhibited by GhCyP3, an interactor of GhPUB17 with antifungal activity. GhCyP3 shows antifungal activity against *V. dahliae*, and the E3 ligase activity of GhPUB17 is repressed by GhCyP3 in vitro [70].

MdPUB29 is a PUB E3 ligase in apple, activated by the infection of fungal pathogen *Botryosphaeria dothidea*. Its overexpression results in enhanced resistance to *B. dothidea* infection in both Arabidopsis and apple calli. Consistent with this, silencing *MdPUB29* resulted in reduced resistance in apple calli [71]. The defense process is accompanied with increased of H_2_O_2_ content and enhanced expressions of SA synthesis- and SA signaling-related genes [71]. The BTB domain is a protein-protein interaction motif in eukaryotes. Many BTB domain-containing proteins play roles as substrate-specific adapters in cullin 3-based E3 ligases [76], and BTB domain proteins have been shown to associate with cullin 3 proteins to form ubiquitin E3 ligases [77]. The apple POZ/BTB Containing-protein 1 (MdPOB1), a BTB-BACK domain E3 ligase, suppresses apple pathogen defense against *B. dothidea* by directly interacting with and degrading MdPUB29 [78].

Oomycete pathogens such as Phytophthora secrete a repertoire of effectors into host cells to manipulate host immunity and benefit infection, including RxLR effectors Avr1b, Avr1d and Avr1k [72]. Several Phytophthora induced GmPUB proteins can interact with the RxLR effectors. Solid data have shown that Avr1b interacts with GmPUB1 in vivo and in vitro. The mutation in Avr1b C-terminus abolishes the interaction with GmPUB1 and the inhibition of cell death. Silencing of *GmPUB1* in soybean cotyledons resulted in loss of recognition by the gene products of Avr1b [72]. Overexpression of *GmPUB1* triggers cell death and enhanced resistance to Phytophthora, revealing that GmPUB1 is a positive regulator of effector-triggered immunity, and plays a positive role in the defense against Phytophthora [72]. Recent data show that Avr1d competes with E2 for GmPUB13 binding to repress the E3 ligase activity and stabilize GmPUB13 to facilitate Phytophthora infection [73]. Thus, GmPUB13 acts as a negative modulator in the infection process. Silencing of *GmPUB13* in soybean hairy roots leads to decreased *P. sojae* infection [73].

NtCMPG1 is highly related to parsley CMPG1 (designated according to the four conserved amino acids, Cys, Met, Pro and Gly) and AtPUB20 and AtPUB21. *NtCMPG1* and the homolog of tomato *SlCmpg1* are induced by Avr9 elicitation. Silencing of *NtCMPG1* leads to reduced HR after Cf-9/Avr9 elicitation, while overexpression of *NtCMPG1* induces a stronger HR in Cf9 tobacco plants post-Avr9 infiltration. In tomato, silencing of *SlCmpg1* decreases the resistance to *Cladosporium fulvum*, while overexpression of epitope-tagged NtCMPG1 mutated in the U-box domain confers a dominant-negative phenotype. Furthermore, *NtCMPG1* is involved in the Pto/AvrPto and Inf1 responses. Therefore, NtCMPG1 is essential for plant defense and disease resistance [79]. CMPG1 is also required for PCD triggered by INF1 (the major elicitin secreted by *P. infestans*) in diverse plant species, and degraded via UPS [80]. The RXLR effector AVR3a of *P. infestans* targets and stabilizes host E3 ligase. In stabilizing CMPG1, AVR3a modifies its normal activity and exclusively suppresses CMPG1-dependent PCD [81], supporting a hypothesis that cytoplasmic effector targets CMPG1 to block the signal transduction and/or regulatory processes following perception of pathogen molecules.

## 6. PUBs Are Multifaceted Modulators in Various Biological Processes

### 6.1. PUBs Control the Development of Roots and Root System Architecture

The root system is a vital plant organ responsible for water absorption, nutrient forage, anchorage, propagation, and storage [82]. Root growth and differentiation are closely linked to plant hormones and shaped by various environmental factors, including nutrient starvation and different stress factors [83]. Compelling data have shown that a variety of PUB proteins are involved in regulating root development (Table 4). AtPUB4 is a novel downstream component of CLV3/CLE signaling pathway in the root meristem [84]. Mutation of *AtPUB4* releases the inhibition of exogenous CLV3/CLE peptide on root cell proliferation and columella stem cell maintenance. Without exogenous CLV3/CLE peptide, the *atpub4* mutants show enhanced root growth, increased cortex/endodermis stem cells and reduced columella layers [85]. Further phenotypic and gene expression assays reveal that AtPUB4 promotes the expression of cell cycle regulatory gene *CYCD6;1*, and regulates formative periclinal asymmetric cell divisions in endodermis and cortex/endodermis initial daughters [85].

AtPUB9 is required for lateral root formation under phosphate starvation conditions. AtPUB9 localizes to autophagic bodies following either activation by S-Domain receptor kinase (ARK2) or under phosphate deficit. The *ark2-1/pub9-1* double mutants display defects in lateral root development and auxin accumulation in the root tips under phosphate deprived conditions [86]. Blocking autophagic responses in WT Arabidopsis results in the inhibition of lateral roots and auxin accumulation in the root tips. It is hypothesized that ARK2/AtPUB9 module might regulate lateral root development using selective autophagy [86].

*TaPUB1* is significantly induced by IAA, and its overexpression results in strengthened photosynthetic capability and increased root length than the non-transgenics. Protein interaction assays show that TaPUB1 directly interacts with and ubiquitinates TaIAA17, facilitating its degradation via UPS, and results in primary root elongation by activating the auxin signaling pathway [45]. Constitutive expression of *TaPUB15* confers more and longer crown root phenotype, and pronounced salt tolerance in rice [49]. Mutation of the rice ortholog *OsPUB15* causes a phenotype of primary root loss and delayed shoot development [87], indicating that both TaPUB15 and OsPUB15 are positive regulators in root development.

Moreover, heteroexpression of *CaPUB1* in Arabidopsis leads to markedly longer hypocotyls and roots and more vigorous growth than WT. Microscopic analysis shows that the roots of *CaPUB1* transgenic plants have more small-sized cells, resulting in disordered, highly populated cell layers in the cortex, endodermis, and stele [41].

### 6.2. PUBs Promote Leaf Senescence via Various Phytohormone Signaling Pathways

Leaf senescence is an active nutrient relocation process regulated by aging, flowering, darkness, nutrient starvation, and environmental stress. Several phytohormones are involved in regulating leaf senescence, including ABA, ethylene, SA and JA. Accumulated data have manifested that PUBs are key modulators in cell senescence via different signaling pathways (Table 4).

Senescence-Associated E3 Ubiquitin Ligase 1 (SAUL1) is a star member in regulating leaf senescence. Its mutation leads to premature senescence under low light conditions in Arabidopsis, with a phenotype of yellow leaves, decreased chlorophyll contents, and accompanied by increased ABA content and enhanced expression of senescence related genes. Aldehyde oxidase 3 (AAO3) is a key enzyme responsible for converting abscisic-aldehyde to ABA [102]. SAUL1 interacts AAO3 in vitro. Mutation of SAUL1 results in the accumulation of AAO3 protein and thus enhances its activity in *saul1* mutants. It is hypothesized that SAUL1 prevents senescence from occurring prematurely by targeting AAO3 for degradation and suppressing ABA production [88].

Transcription factor ORESARA1 (ORE1) is a key regulator of age-dependent leaf senescence and cell death in Arabidopsis. The accumulation of ORE1 transcripts is induced by low light treatment in *saul1* mutants, along with the enhanced expression of senescence marker gene *Senescence-Associated-Gene 12* (*SAG12*) and cell death. However, the accumulation of ORE1 is not sufficient to cause *saul1* phenotypes, as demonstrated by double mutant analysis. Exposure of *saul1* mutants to low light for 24 h does not cause visible senescence symptoms; however, the senescence-promoting TF genes *WRKY53*, *WRKY6*, and *NAC-LIKE ACTIVATED BY AP3/PI* are up-regulated, indicating the initiation of senescence in *saul1* seedlings. Transcriptomic data reveal that SA might play a central role in *saul1* senescence and cell death, which is verified by increased SA content in low-light-treated *saul1* mutants, and application of exogenous SA triggers *saul1* senescence in permissive light conditions. Furthermore, *saul1* senescence depends on the PAD4-dependent SA pathway [103].

*AtUSR1* (*U-box Senescence Related 1*) encodes a PUB protein. Its transcription is promoted by MYC2 TF in JA signaling pathway. Overexpression of *AtUSR1* in the *myc2* mutant plants show precocious senescence, while *myc2* mutant displays a phenotype of delayed leaf senescence, suggesting that AtUSR1 functions downstream to MYC2 in the JA signaling pathway in promoting leaf senescence [89].

Additionally, AtPUB13 transgenic plants and the *pub12pub13* mutant are more sensitive to stress-induced leaf senescence accompanied by elevated expression of stress-induced senescence marker genes [104]. Therefore, PUB genes are widely involved in the process of senescence.

### 6.3. PUBs Are Required in Controlling Flowering Time

AtPUB13 is an essential flowering regulator in Arabidopsis. The *atpub13* mutant shows a phenotype of early flowering under middle- and long-day conditions, in which the expressions of *SOC1* (Suppressor of Overexpression of Constants 1) and Flowering Locus T (*FLT*) are induced, while Flowering Locus C (*FLC*) expression is suppressed. The two components involved in the SA-mediated signaling pathway, SID2 and PAD4, are required for flowering-time control. Briefly, AtPUB13 acts as an important node connecting SA-dependent defense signaling and flowering-time regulation in Arabidopsis [61].

OsSPL11, a homolog of AtPUB13 in rice, employs a different pathway to control flowering time, by interacting with the novel RNA/DNA binding KH domain protein SPIN1. Overexpression of *SPIN1* causes late flowering in transgenic rice under both short- and long-day conditions. RNA-binding and SPIN1-interacting 1 (RBS1) protein binds RNA in vitro and interacts with SPIN1 in the nucleus. *RBS1* overexpression represses the expression of *Hd3a* and causes delayed flowering under both short- and long-day conditions [90]. Interestingly, the expression of *SPIN1* is increased, while *OsSPL11* is repressed in the *RBS1* overexpression plants. Moreover, *RBS1* is up-regulated in both *SPIN1* overexpressing plants and the *osspl11* mutant. Collectively, OsSPL11 modulates flowering time by interacting with SPIN1, the positive regulator of RBS1; while RBS1 inhibits flowering by represses the expression of *Hd3a* in rice [90].

In addition, GmPUB8 participates in controlling flowering by interacting with COL (Constans Like) protein in soybean. *GmPUB8-overexpressing* Arabidopsis plants show an early flowering phenotype under middle- and short-day conditions, but delayed flowering under long-day conditions, indicating that GmPUB8 might regulate flowering time via the photoperiod pathway [17].

### 6.4. PUBs Participate in Fruit Development and Ripening via the Phytohormone-Dependent Pathways

Fruit ripening is a complicated process involving different phytohormones and various environmental stimuli. Fruit were classified into climacteric and nonclimacteric based on the patterns of respiration and ethylene production during fruit maturation and ripening [105]. Accumulated data have shown that PUBs are involved in the ripen of both climacteric and nonclimacteric fruit (Table 4).

Apple MdbHLH3 is an anthocyanin-related basic helix-loop-helix TF. It binds to the promoters of ethylene biosynthesis genes *MdACO1*, *MdACS1*, and *MdACS5A*, activating their transcriptions and promoting ethylene biosynthesis. Overexpression of *MdbHLH3* leads to the enhancement of ethylene production, premature leaf senescence, and apple fruit ripening [91]. MdPUB29 can directly ubiquitinate MdbHLH3 guiding its degradation via UPS, therefore inhibit the expressions of ethylene biosynthetic genes, ethylene production and apple fruit ripen, demonstrating that MdPUB29 is implicated in controlling climacteric fruit ripen [91].

Recent studies have revealed that PUBs also involved in the ripen of nonclimacteric fruit. For instance, VlPUB38 is a PUB member in grape. Its overexpression results in delayed mature in strawberry. However, this phenotype can be rescued by applying exogenous ABA and the inhibitor of 26S proteasome, MG132. Furthermore, VlPUB38 interacts with abscisic-aldehyde oxidase (VlAAO), targeting VlAAO proteolysis via UPS. Briefly, VlPUB38 negatively modulates grapefruit ripening by mediating the degradation of the key component VlAAO in the ABA synthesis pathway [92].

Moreover, transcriptome profiling also shows that the expression of 37–40 PUB genes in the pre-ripen and ripen peach fruit, hinting the intensive involvement of PUBs in fruit development and ripen [22].

### 6.5. Multiple PUBs Are Key Regulators of Sexual Reproduction in Flowering Plants

Accumulated data have shown that PUB E3 ligases are vital modulators in the sexual reproduction of flowering plants, including male sterility and self-incompatible (Table 4). Male sterility is generally characterized by the impairment of the male reproductive development as a result of underlying genetic causes and leads to the malformation of male gametes and/or pollen, or unable release. Pollen formation is a complex developmental process that has been extensively investigated. AtPUB4, a U-box/ARM repeat-containing E3 ligase, is a novel player in male fertility in Arabidopsis. Loss function of AtPUB4 causes hypertrophic growth of the tapetum layer, incomplete degeneration of tapetal cells and strikingly abnormal exine structures of pollen grains. Although the *atpub4* mutant produces viable pollen, the pollen grains adhere to each other and to the remnants of incompletely degenerated tapetal cells, and do not properly disperse from dehisced anthers for successful pollination. Further studies indicates that the mutation caused sterile is temperature-dependent. The *atpub4* mutant plants are completely sterile at 22 °C, but partially fertile at 16 °C [93]. Recent research has shown that AtPUB4, along with AtPUB2, interacts with Extra-Large G Proteins (XLGs) and might function in the complex cytokinin-signaling networks. The *atpub4* mutant, the *atpub2/4* double mutant, and *atxlg1/2/3* triple mutant all exhibits defects in cytokinin responses, stamen and tapetum development, and male fertility [94]. Overexpressing *ARR10*, a positive regulator gene of cytokinin signaling, to enhance the cytokinin response in *atpub4* or in the *atxlg1/2/3* triple mutant partially restore several phenotypes caused by the *atpub4* and *atxlg1/2/3* mutations [94].

In rice, the *ospub73* mutant displayed significant lower pollen fertility (19.45%) relative to WT (85.37%). Cytological evidence shows that the tapetum of *ospub73* mutant is vacuolated at the meiosis stage, and the pollen exine is abnormal at the bi-cellular pollen stage. Transcriptomic assays show that seven known genes associated with tapetal cell death or pollen exine development are down-regulated, including *CYP703A3* (Cytochrome P450 Hydroxylase703A3), *CYP704B2* (Cytochrome P450 Hydroxylase704B2), *DPW* (Defective Pollen Wall), *PTC1* (Persistent Tapetal Cell1), *UDT1* (Undeveloped Tapetum1), *OsAP37* (Aspartic Protease37) and *OsABCG15* (ATP Binding Cassette G15), suggesting that OsPUB73 play an important role in tapetal or pollen exine development [95].

Self-incompatibility (SI) is one of the most important mechanisms to prevent self-fertilization and generate genetic diversity within flowering plant species. The SI response is comprised of a self- and non-self-recognition process between pollen and pistil following a selective inhibition of the self-pollen (tube) development. The self-/non-self-recognition in most species is controlled by a single multiallelic S-locus. Pollen inhibition occurs when the same “S-allele” specificity is expressed by both pollen and pistil [106]. In the Brassicaceae, the self-pollen is recognized by the pollen S-locus cysteine rich/S-locus protein 11 (SCR/SP11) ligand and the pistil S receptor kinase (SRK). The SCR/SP11 ligand on the pollen surface binds to SRK on the pistil surface, and initiate the SRK-activated signaling pathway. The self-pollen is rejected by preventing pollen hydration following pollen contact with the stigmatic surface. The armadillo repeat-containing 1 (ARC1) protein, a member of the PUB E3 ubiquitin ligases, is involved in this signaling pathway. ARC1 is required downstream of SRK for the self-incompatibility response [96,97]. ARC1 can shuttle between the nucleus, cytosol, and proteasome/COP9 signalosome in tobacco BY-2 cells. However, ARC1 localization to the proteasome/CSN occurs only in the presence of an active SRK. Inhibition of the proteasomal proteolytic activity disrupts the SI response. Thus, ARC1 might propel the ubiquitination and proteasomal degradation of compatibility factors in the pistil, causing pollen rejection [97]. Subsequent studies have shown the deletion of ARC1 in many self-mating plant species, demonstrating that the conserved role for ARC1 in the self-pollen rejection response within the Brassicaceae [107].

### 6.6. PUBs Are Involved in Regulating Cell Division

*Dwarf and Short Grain 1* (*DSG1*) encodes a nuclear- and cytoplasm-localized PUB E3 ligase in rice. Mutation of *DSG1* cause a phenotype with shorter roots, internodes, panicles and seeds, and wider and curled leaves, due to defects in cell division and elongation [98]. Furthermore, the *dsg1* mutant is less sensitive to BRs, and *DSG1* expression is negatively regulated by BRs, ethylene, auxin, and SA. Collectively, DSG1 positively regulates cell division and elongation and is involved in multiple hormone pathways [98]. Additionally, AtPUB4 regulates asymmetric cell division and cell proliferation in the root meristem [85].

### 6.7. PUBs Modulate Nodulation in Leguminous Plants

The legume-rhizobia symbiosis is initiated by a signal exchange between host plants and soil bacteria. Root nodulation requires the dual activation of nodule organogenesis and infection processes, which depend on the perception of Rhizobial Nod factor (NFs), lipochitooligosaccharadic signals in the plant roots. Lumpy infections (LIN) is a PUB-type E3 ligase in *Medicago truncatula*, functions at an early stage of the Rhizobial symbiotic process, and is required for both infection thread growth in root hair cells and the further development of nodule primordia, mutation of *LIN* leads to a suppression of nodule development [99]. Receptor-like kinases (RLKs) play multiple roles in the perception of NFs and their transduction via calcium-mediated responses and transcriptional regulation [108]. LYK3 is a RLK of *M. truncatula*, which is essential for the establishment of the nitrogen-fixing, root nodule symbiosis with *Sinorhizobium meliloti*. MtPUB1 is an interactor of the kinase domain in LYK3. Both are localized and interact in the plasma membrane of plants. *MtPUB1* is induced by NFs and expresses specifically in symbiotic conditions, and shows an overlapping expression pattern with LYK3 during nodulation. MtPUB1 has ubiquitin ligase activity and is phosphorylated by the LYK3. Overexpression of *MtPUB1* leads to a delay in nodulation with *S. meliloti*, suggesting that it is a negative regulator of nodulation [100]. Furthermore, the transcript level of *MtPUB1* is a specificity determinant of nodulation to Rhizobial species in a NF-dependent manner [100].

Does Not Make Infection 2 (MtDMI2) is a Leu rich repeat-type receptor kinase required for signal transduction in the *M. truncatula*/*S. meliloti* symbiosis pathway. MtPUB2 is a novel PUB-type E3 ligase, and can be phosphorylated at Ser-316, Ser-421, and Thr-488 residues by MtDMI2. The phosphorylation status of Ser421 is essential for the ubiquitination activity of MtPUB2. MtDMI2 can be persistently ubiquitinated by MtPUB2S421D (mimics the phosphorylated state) and degraded via UPS. However, the mutant MtPUB2S421A loses the ubiquitination function [101]. Further studies reveal that MtDMI2-MtPUB2 form a prey-predator type negative feedback loop to maintain the nodulation homeostasis [101].

## 7. Concluding Remarks and Perspectives

In recent decades, tremendous progress has been made to understand the roles of PUB E3 ubiquitin ligases in plants. Many PUBs are found in various plant species. PUBs are essential regulators in gene expression by controlling MiRNA biogenesis and involved in various phytohormone signaling pathways. The multifarious functions of PUB E3 ligases can be divided into four categories according to their roles in various biological processes: (1) determination of sexual reproduction; (2) modulating plant growth and development; (3) participation in the signaling of biotic stress and abiotic stress (Figure 1); and (4) most importantly, serving as regulators in the response to various environmental stresses.

Climate change is associated with increased CO_2_ concentration and has led to shifts in temperature and rainfall patterns [109], resulting in myriad biotic and abiotic constraints to crop production. Given how crops respond to various environmental stresses, PUBs could represent a gene resource to develop more climate-resilient crops [110]. Nevertheless, it should be pointed out that several PUB members play completely opposite roles in different stresses; for instance, CaPUB1 plays a positive role in cold tolerance, but acts as a negative modulator in salt and drought tolerance [40]. Only very few PUB members play positive roles in response to multiple environmental stresses (Table 2, Table 3 and Table 4). In addition, there is a lack of comprehensive evaluation of phenotypic effects for most PUBs, especially on agronomic traits, which should be addressed in future studies.

Phylogenetic data show that the PUB gene family has undergone different duplication events in the process of evolution, leading to the emergence of new members, especially in polyploidy plant species. Up to now, only very few PUB members have been structurally characterized. To gain a deeper understanding of PUB’s function, more attention should be paid to crystal structural analysis to decipher the interactions with upstream regulators and downstream interactors or substrates.

Additionally, among the identified PUB proteins, some are well characterized. However, most of these studies are preliminary (Table 2, Table 3 and Table 4). The upstream regulators and downstream substrates, as well as the regulatory networks, are largely unknown. To reveal how many faces PUBs have, research in this area need to be strengthened. A more comprehensive and in-depth study will help us to understand its functional diversity and promote its application in crop genetic improvement.

## Figures and Tables

**Figure 1 ijms-23-02285-f001:**
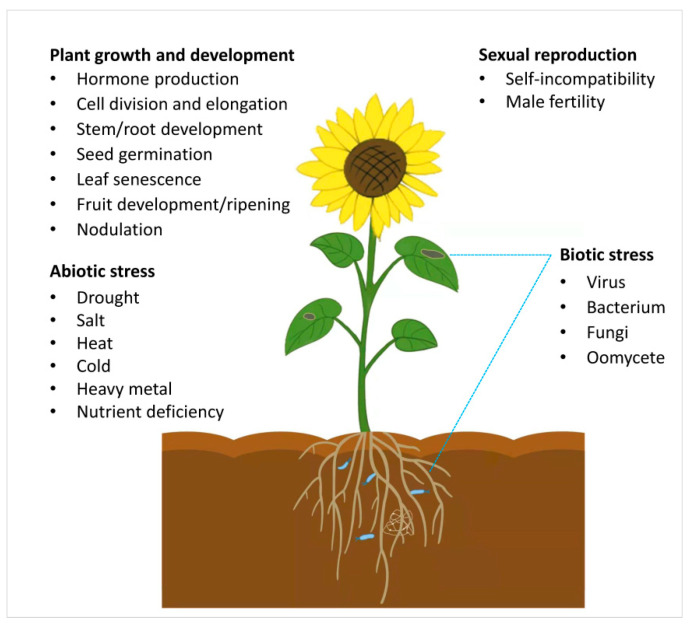
The roles of PUBs in plant growth and development and response to biotic and abiotic stress.

**Table 1 ijms-23-02285-t001:** Comparisons of gene number and classifications of the U-box E3 ubiquitin ligases in various species.

Species	Gene Number	Classifications	References
*Saccharomyces cerevisiae* (Yeast)	2	2	[10]
*Homo sapiens* (Human)	8	7	[10]
*Chlamydomonas reinhardtii*	30	3	[11]
*Arabidopsis thaliana*	64	8	[2]
*Oryza sativa* (Rice)	77	8	[2]
*Horderm vulgare* (Barley)	67	8	[13]
*Triticum turgidum* ssp. *Dicoccoides* (Emmer wheat)	82	7	[19]
*Triticum aestivum* (Bread wheat)	213	7	[15]
*Glycine max* (Soybean)	125	8	[17]
*Brassica oleracea* (Cabbage)	99	7	[20]
*Brassica rapa* ssp. *Pekinesis* (Chinese cabbage)	101	10	[21]
*Lycopersicon esculentum* (Tomato)	62	4	[14]
*Musa nana* (Banana)	91	6	[20]
*Prunus persica* (Peach)	54	7	[22]
*Pyrus bretschneideri* (Pear)	62	5	[23]
*Gossypium arboretum* (Diploid cotton)	96	11	[16]
*G. barbadense* (Tetraploid cotton)	208	14	[16]
*G. hirsutum* (Tetraploid cotton)	185	17	[16]
*G. raimondii* (Diploid cotton)	93	11	[16]

**Table 2 ijms-23-02285-t002:** PUBs Function as Positive or Negative Regulators in the Response to Abiotic Stress.

Gene	Abiotic Stress	Biological Roles	Upstream Regulator	Interactor/ Substrate	Transgenic Plants	References
*AtPUB11*	ABA, drought	Negative	-	LRR1, KIN7	Arabidopsis,	[32]
*AtPUB18*, *AtPUB19*	ABA, drought	Negative	-	Exo70B1	Arabidopsis	[33]
*AtPUB2*, *AtPUB23*	Drought	Negative	-	RPN12a, Exo70B2	Arabidopsis	[34,35]
*OsPUB41*	ABA, drought	Negative	OsUBC25	OsCLC6	Rice	[37]
*OsPUB67*	Drought	Positive	-	OsRZFP34, OsDIS1	Rice	[38]
*GmPUB6*, *GmPUB8*	Drought	Negative	-	-	Arabidopsis	[17,39]
*StPUB27*	Drought	Negative	-	-	Potato	[32]
*CaPUB1*	Drought, salt	Negative	-	-	Arabidopsis, rice	[40,41]
	Cold	Positive	-	-	Rice	[40]
*TaPUB1*	Drought, salt	Positive	-	-	Arabidopsis, wheat, tobacco	[42,43,44]
	Cd, IAA	Positive	-	TaIRT1, TaIAA17	Wheat	[45]
*PbrPUB18*	Drought	Positive	-	-	Arabidopsis	[23]
*PalPUB79*	Drought	Positive	PalWRKY77	PalWRKY77	Poplars	[46]
*PnSAG1*	ABA, salt	Negative	-	-	Arabidopsis, *Physcomitrella patens*	[47]
*AtPUB30*	Salt	Negative	-	BKI1	Arabidopsis	[48]
*TaPUB15*	Salt	Positive	-	-	Arabidopsis	[49]
*TaPUB26*	Salt	Negative	-	-	Wheat, Brachypodium	[50]
*AtPUB48*	Heat	Positive	-	-	Arabidopsis	[51]
*SlCHIP*	Heat	Positive	-	-	Tomato	[52]
*AtPUB25*, *AtPUB26*	Cold	Positive	AtOST1	AtMYB15	Arabidopsis	[53]
*VpPUB24*	Cold	Positive	VpICE1	VpHOS1	Arabidopsis	[54,55]
*CrPUB5 CrPU14*	Nitrogen starvation	Negative	-	-	*C. reinhardtii*	[11]
*CrPUB11, CrPUB23, CrPUB28*	Nitrogen starvation	Positive	-	-	*C. reinhardtii*	[11]

-, indicates unknown.

**Table 3 ijms-23-02285-t003:** PUBs Involved in the Response to Pathogen Attacks.

Gene	Pathogen Factors	Biological Roles	Upstream Regulator	Interactor/Substrate	Transgenic Plants	References	
*AtPUB12, AtPUB13*	Bacteria	Negative	BAK1	FLS2	Arabidopsis	[26,60]	
*AtPUB25, AtPUB26*	*Verticillium dahliae*	Negative	-	AtMYB6	Arabidopsis	[63]	
*OsSPL11*	*Xanthomonas oryzae*,	Negative	-	SPIN6	Rice	[64,65]	
*OsPUB15*	*M. oryzae*	Positive	Pid2	-	Rice	[66]	
*OsPUB44*	*M. oryzae*	Positive	-	XopP	Rice	[67]	
*P3IP1*	Rice grassy stunt virus	Negative	-	OsNRPD1a	Rice	[68]	
*StPUB17*	*Phytophthora infestans*	Positive	-	-	Potato	[69]	
*GhPUB17*	*V. dahliae*	Negative	GhCyP3	-	Cotton	[70]	
*MdPUB29*	*Botryosphaeria dothidea*	Positive	MdPOB1	-	Arabidopsis, apple calli	[71]	
*GmPUB1*	*Phytophthora sojae*	Positive	-	-	Soybean	[72]	
*GmPUB13*	*P. sojae*	Negative	-	-	Soybean	[73]	

-, indicates unknown.

**Table 4 ijms-23-02285-t004:** PUBs Participate in Plant Growth and Development.

Gene	Plant Traits	Biological Roles	Upstream Regulator	Interactor/ Substrate	Transgenic Plants	References
*AtPUB9*	Lateral root	Positive	-	-	Arabidopsis	[86]
*TaPUB1*	IAA, root development	Positive	-	TaIAA17	Wheat	[45]
*TaPUB15*	Root development	Positive	-	-	Rice	[49]
*OsPUB15*	Root and shoot development	Positive	-	-	Rice	[87]
*CaPUB1*	Root development	Positive	-	-	Arabidopsis	[41]
*GmPUB6*	Seed germination, root development	Negative	-	-	Arabidopsis	[39]
*SAUL1*	Leaf senescence	Positive	-	AAO3	Arabidopsis	[88]
*AtUSR1*	Leaf senescence	Positive	MYC2	-	Arabidopsis	[89]
*AtPUB13*	Flowering	Negative	-	-	Arabidopsis	[61]
*OsSPL11*	Flowering	Positive	-	RBS1	Rice	[90]
*GmPUB8*	Flowering	Positive	-	-	Arabidopsis	[17]
*MdPUB29*	Ethylene production, Fruit ripening, leaf senescence	Positive	MdbHLH3	-	Apple	[91]
*VlPUB38*	Fruit ripening	Negative	-	VlAAO	Grape	[92]
*AtPUB4*	Tamen development, male fertility,	Positive	-	-	Arabidopsis	[93,94]
	cell division	Negative	-	-	Arabidopsis	[85]
*OsPUB73*	Tapetal and pollen exine development	Positive	-	-	Rice	[95]
*ARC1*	Self-incompatibility	Positive	SRK	-	Brassica	[96,97]
*DSG1*	cell division and elongation	Positive	-	-	Rice	[98]
*LIN*	nodule primordium development	Positive	-	-	*Medicago truncatula*, *Lotus japonicus*	[99]
*MtPUB1*	Nodulation	Negative	LYK3	-	*M. truncatula*	[100]
*MtPUB2*	Nodulation homeostasis	Negative	MtDMI2	-	*M. truncatula*	[101]

-, indicates unknown.

## Data Availability

Not applicable.

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
