# Peer review of "How Many Faces Does the Plant U-Box E3 Ligase Have?"

_ijms, 2022, doi:10.3390/ijms23042285_

Round 1

Reviewer 1 Report

Dear Authors,

The manuscript entitled "How many faces does the plant U-box E3 ligase have?" is a well-thought-out and written review paper. Of course, more can always be written. The work has a logical layout. I would suggest swapping sections 6.1. and 6.2. and 6.5. and 6.6.

At the beginning of the introduction, there is a piece of text leftover from the template. When you give the Latin name of the species for the first time, use the full name together with the authority. 

Author Response

Point 1: I would suggest swapping sections 6.1. and 6.2. and 6.5. and 6.6.

Response 1: We swapped the sections according to the reviewer’s suggstion.

Point 2: At the beginning of the introduction, there is a piece of text leftover from the template. When you give the Latin name of the species for the first time, use the full name together with the authority. 

Response 2: We removed the template and revised the Latin names of species acorrdingly.

Reviewer 2 Report

I don't have any further comments on this manuscript since it is well written and easy to understand. As a person not familiar with PUB genes, the manuscript offers much information regarding the speciality of PUBs. There are only a few minor errors that need to be revised. 

  1. The first paragraph is the comment from a reviewer? It is not the context.
  2. Line108, this is an error. 
  3. Line 768 has a similar error. 
  4. I would like to suggest the authors produce a schematic that PUBs are involved in abiotic and biotic stress tolerance. 

Author Response

Response to Reviewer 2 Comments

Point 1: The first paragraph is the comment from a reviewer? It is not the context.

Response 1: Sorry, it is the template. We removed it from the revised MS.

Point 2: Line108, this is an error. 

Response 2: The error was deleted.

Point 3: Line 768 has a similar error. 

Response 3: The error was removed.

Point 4: I would like to suggest the authors produce a schematic that PUBs are involved in abiotic and biotic stress tolerance. 

Response 4: We added a schematic in the conclusion section.
